# Enhancing Radiotherapy Workflow for Head and Neck Cancer with Artificial Intelligence: A Systematic Review

**DOI:** 10.3390/jpm13060946

**Published:** 2023-06-02

**Authors:** Ciro Franzese, Damiano Dei, Nicola Lambri, Maria Ausilia Teriaca, Marco Badalamenti, Leonardo Crespi, Stefano Tomatis, Daniele Loiacono, Pietro Mancosu, Marta Scorsetti

**Affiliations:** 1Department of Biomedical Sciences, Humanitas University, via Rita Levi Montalcini 4, Pieve Emanuele, 20072 Milan, Italy; 2IRCCS Humanitas Research Hospital, Radiotherapy and Radiosurgery Department, via Manzoni 56, Rozzano, 20089 Milan, Italy; 3Dipartimento di Elettronica, Informazione e Bioingegneria, Politecnico di Milano, 20133 Milan, Italy; 4Centre for Health Data Science, Human Technopole, 20157 Milan, Italy

**Keywords:** radiotherapy (RT), head and neck cancer (HNC), artificial intelligence (AI), machine learning (ML), deep learning (DL), automation

## Abstract

Background: Head and neck cancer (HNC) is characterized by complex-shaped tumors and numerous organs at risk (OARs), inducing challenging radiotherapy (RT) planning, optimization, and delivery. In this review, we provided a thorough description of the applications of artificial intelligence (AI) tools in the HNC RT process. Methods: The PubMed database was queried, and a total of 168 articles (2016–2022) were screened by a group of experts in radiation oncology. The group selected 62 articles, which were subdivided into three categories, representing the whole RT workflow: (i) target and OAR contouring, (ii) planning, and (iii) delivery. Results: The majority of the selected studies focused on the OARs segmentation process. Overall, the performance of AI models was evaluated using standard metrics, while limited research was found on how the introduction of AI could impact clinical outcomes. Additionally, papers usually lacked information about the confidence level associated with the predictions made by the AI models. Conclusions: AI represents a promising tool to automate the RT workflow for the complex field of HNC treatment. To ensure that the development of AI technologies in RT is effectively aligned with clinical needs, we suggest conducting future studies within interdisciplinary groups, including clinicians and computer scientists.

## 1. Introduction

Head and neck cancer (HNC) represents a heterogeneous and challenging group of tumors for which radiotherapy (RT) is considered an effective treatment in the multidisciplinary management of both early and locally advanced stages [1]. Currently, the RT workflow for HNC patients is time consuming and laborious, as it involves a region characterized by complex-shaped tumors and numerous organs at risk (OARs), inducing challenging RT planning and delivery.

Standardization, efficiency, and consistency in treatment are constantly sought after by clinicians and researchers employed in RT. In the last few years, artificial intelligence (AI) has been tested in the RT field to streamline and improve processes [2,3]. AI is usually defined as the discipline, heavily based on mathematics, statistics, and computer science, devoted to the study and design of methods that are able to mimic human intelligence, such as visual recognition and problem solving. The increasing availability of large amounts of data and computational resources in the last two decades has enabled the rise of machine learning (ML), a subset of AI that consists of classes of algorithms, statistical models, and mathematical models capable of learning how to perform specific tasks by learning the representation of raw data without manual feature engineering. In recent years, another class of methods referred to as deep learning (DL) has attracted a lot of interest due to the extremely competitive performances shown on many different tasks. DL is a subset of ML, comprising complex models relying on artificial neural networks. So far, these methods have proven successful on several applications involving high dimensional data such as images and text, for which it is often difficult to extract a set of meaningful features.

In the medical field, and in particular in RT, such methods have been tested for the possibility of modeling dose-response relations thanks to the integration of medical imaging and clinical features. ML/DL methodologies may be useful to identify tumor heterogeneity and intrinsic radioresistance or to evaluate normal tissue responsiveness to radiation [4]. Among several applications, ML/DL has been rapidly adopted to improve the RT workflow for HNC [5,6,7,8,9].

To the best of our knowledge, few reviews have focused on the applications of AI for HNC treatment in RT, including target and OAR segmentation [10,11], radiomics [12], and the HNC planning process [13]. In this article, we have provided a review of the use of AI in HNC patients according to the whole clinical workflow of RT, which includes contouring, planning, and delivery phases.

## 2. Materials and Methods

The “Pubmed” database was queried to search for English papers published or in-press (i.e., online first) between 1 January 2016 and 1 October 2022. Articles published without external review (e.g., arXiv) or reviewed but not yet online were not taken into consideration. Original papers that contained in the title or in the abstract the following words were considered: (“Head” and “Neck” or “pharynx” or “oral cavity” or “larynx”) and (“Radiation Oncology” or “radiotherapy”) and (“artificial intelligence” or “machine learning” or “deep learning”). Systematic reviews, meta-analyses, case reports, non-English language scientific articles, books, and papers that did not rely on a medical image dataset were not considered.

Each selected paper was individually assessed by two expert radiation oncologists by considering relevance to the topic, novelty, methodology, and quality of evidence. In the case of discrepancies, a third expert resolved the conflict. The papers were divided into these three sections: (1) “AI in HNC target and organs at risk contouring”; (2) “AI in HNC RT planning optimization”; (3) “AI during the HNC RT delivery”. Each section reports a descriptive summary with an exhaustive table presenting the AI technologies tested and the outcomes obtained, as well as a dedicated subsection on the future prospects for the research area. Papers focusing on radiomics and outcome prediction were not included in this review.

## 3. Results

A total of 168 scientific articles were screened through PubMed in blind mode by two radiation oncologists, who agreed on 66 papers. A total of 12 out of 16 paper conflicts were accepted by a third radiation oncologist, for a total of 78 articles. The articles were further revised to consider only relevant studies. Eventually, the group agreed on 62 papers that were subdivided into the three sections mentioned above. The count of selected articles per group is shown in Figure 1.

### 3.1. AI in HNC Target and Organs at Risk Segmentation

Contouring is a crucial step in HNC RT. The location of critical structures and the clinical target volumes (CTV) involving the disease site and prophylactic areas make HNC challenging and, thus, prone to errors and variability between operators. Guidelines are essential to defining treatment volumes and OARs, promoting harmonization, and minimizing contouring variability. However, even with guidelines in place, the variability between different observers is not negligible. Therefore, contouring automation could enhance treatment homogeneity and, additionally, make the workload more manageable, as this phase of the workflow is extremely time-consuming. A lot of interest in this field has been growing over the past few years with the introduction of several methodologies, including atlas-based (AB) and convolutional neural networks (CNN), to increase the accuracy of segmentation [14,15]. Appendix A summarizes the findings of the studies focusing on target and/or OAR segmentation.

The first studies that paved the way for the explosion of research in AI applied to HNC segmentation were those of Ibragimov et al. [16] and Tam et al. [17], who demonstrated the feasibility and effectiveness of ML/DL models for the segmentation of OARs in HNC patients. For the models’ performance evaluation, the most commonly used scores were the Dice Similarity Coefficient (DSC), Hausdorff distance 95% (HD95), and Mean Surface Distance (MSD). DSC measures the overlap degree between regions, typically the manually delineated contours (“ground truth”) and the automatically generated segmentation. HD95 calculates the 95th percentile of point distances, while MSD measures the average distance between surfaces. Nikolov et al. introduced the surface DSC, which quantifies the deviation between surface contours rather than volumes [18]. Furthermore, the authors proposed and tested a 3D U-Net model on external public datasets, demonstrating adequate generalizability.

Most of the published studies evaluated the role of segmentation in OARs rather than the tumor. One of the biggest real-world data experiences was published by Zhong et al., in which the OARs of 664 HNC patients were segmented using a CNN, showing good results (DSC > 0.7) for many OARs [19]. Zhang et al. developed an automatic tool with a total segmentation time of 40 seconds [20], while Brunenberg et al. reported variable performances, with more adequate segmentations for the glandular OARs, brainstem, mandible, and oral cavity than for other aero-digestive tracts [21].

Many authors compared DL to AB, showing heterogeneous results. Chen et al. demonstrated significant better performance of a 3D U-Net vs. AB [22]. Van Dijk et al. [23] compared CNN to AB segmentations of 589 simulation CT scans for 22 manually contoured OARs. A significantly better performance of CNN on 19/22 structures was observed, with a large reduction in the overall delineation time compared to AB. On the contrary, Urago et al. [24] found no significant difference between the two techniques. Moreover, Guo et al. showed that DL for OAR segmentation had no significant impact on dose-volume metrics [25].

A modified fully connected DenseNet (FC-DenseNet) was employed by Kim et al. for DL segmentation of 23 OARs [26]. The authors trained two FC-DenseNet models using matched (DLSm) and unmatched (DLSu) patients in the test set and compared the models’ performance with deformable image registration (DIR). Overall, DLSm achieved better outcomes than both DLSu and DIR, mainly for glandular structures. Twenty-six observers subjectively evaluated 9 OARs: DLS was perceived as a human in 49% of cases, whereas DLSm was preferred over DLSu (67%) and DIR (97%), with a similar rate of required revision (28% vs. 30%, respectively).

Several authors investigated the integration of manual adjustments into OARs after automatic contouring. Brouwer et al. reported that the median manual adjustment was lower than 2 mm for all structures [27]. Oktay et al. reported a 93% time reduction for the auto-segmented OARs, with a further manual correction of ~5.0 min/scan [28]. A semi-automated review process was tested by Bai et al. [29], in which a one-click procedure allowed an iterative update of the proposed model. The authors showed the generated contours improved in DSC (>10%) and HD95 (almost halved) after only three updates. Finally, Wong et al. demonstrated that most OARs required minimal edits after automatic segmentation [30]. 

A specific study on small OARs was performed by Liu et al. for nasopharyngeal tumors [31]. The authors validated their model against other state-of-the-art networks on the public StructSeg 2019 challenge dataset, reporting an average DSC = 0.80.

Another area of investigation was the use of a three-dimensional multi-view model (coronal, sagittal, and transverse planes). While with this approach Liu et al. obtained a minimal DSC improvement (0.83 vs. 0.86) [32], Iyer et al. trained a DeepLabV3+-based model on multi-view CTs and obtained better segmentations in the case of occasional single-view model failures [33]. Other studies have been published on the same topic by Wong et al. and Zhang et al. [34,35], but they exhibited limited clinical impact.

The combination of multiple DL networks in parallel or cascade has also been tested with interesting results. A shape representation model and a CNN were proposed by Tong et al. [36], while Liang et al. adopted two CNNs in series to (i) delimit organ bounding boxes and (ii) predict segmentation masks for each organ within the bounding boxes [37]. The authors achieved a mean time of 9.5 s to segment 9 OARs. Similarly, Men et al. [38] and Zhong et al. [39] presented multiple cascaded CNNs. Tappeiner et al. showed that a DL-based approach composed of two networks, one dedicated to the gross recognition of the regions of interest (ROIs) and the other capable of producing a fine segmentation, performed better than atlases and model-based methods [40]. Similar findings were reported by Sultana et al., using (i) a U-Net for the recognition of the ROIs and (ii) a generative adversarial network (GAN) for the segmentation [41]. The segmentation of the parotid gland was investigated by Hänsch et al. [42] by comparing three different types of networks. No significant differences were observed.

The size of the training data in terms of the number of scans might affect the accuracy of a segmentation network. According to Tappeiner et al., a number of at least 12 images is sufficient for accurate auto-segmentation, with only a 3% drop for OARs compared to a full set of 25 images [43]. A training sample size of 200 patients was reported by Fang et al. to have the 95% best effect on optic nerves and lenses, while 40 patients were necessary for other OARs [44].

Most of the studies evaluated only CT-based segmentation approaches, but some groups also tested models for magnetic resonance imaging (MRI) segmentation. Hague et al. reported significantly better results with an MRI-based model for parotid glands and submandibular glands [45]. Dai et al. trained a CNN model to extract features from MRI sequences, obtaining a mean DSC = 0.78 [46]. Korte et al. investigated DL-based auto-segmentation on MRI, demonstrating good results for parotid and submandibular glands [47]. Dai et al. [48] and Kieselmann et al. [49] proposed, respectively, a Cycle-GAN and a CNN to segment OARs on synthetic MRIs generated from CT images.

Studies on AI-based gross tumor volume (GTV) delineation were typically performed using mainly PET and CT. Comelli et al. developed a segmentation algorithm for reconstructing the 3D shape of the tumor on PET data [50]. Naser et al. utilized both PET and CT series simultaneously using 2D and 3D models [51]. Groendahl et al. compared manual GTV delineation to the conventional PET threshold method, classical ML, and a PET/CT-based CNN [52]. The PET/CT-based CNN model achieved significantly better segmentation. Guo et al. used a Dense-Net architecture to segment GTV in HNC patients with heterogeneous disease sites [53]. Their method used features extracted from both planning CT and PET scans. Gurney-Champion et al. achieved a mean DSC score of 0.87 using a 3D U-Net model for segmenting primary tumors and lymph node metastases on diffusion-weighted MRI [54]. Ren et al. trained a residual 3D U-Net on four different imaging combinations (CT-PET-MRI, CT-MRI, CT-PET, and PET-MRI) to contour the tumor and the involved lymph nodes [55]. The inclusion of the PET image was crucial to achieving acceptable model performance, and the CT-MRI combination provided poorer results than all other combinations that included PET images. Moe et al. compared CT-based, PET-based, and PET/CT-based CNN models for GTV contouring, achieving a mean DSC of 0.55, 0.69, and 0.71, respectively [56].

Lastly, the inter-observer variability in structure contouring significantly limits the delineation accuracy and may have an impact on the interpretation of outcomes from multicenter trials [57,58]. Van der Veen et al. developed a CNN capable of creating each neck nodal level separately, which allowed them to compare the time required to correct the automatic segmentation with the time required to produce it de novo [59]. A clear advantage of the automatic mode was demonstrated, with 35 vs. 52 min for whole levels and 8 vs. 15 min when only target levels related to a clinical plan were selected. Furthermore, the inter-observer variability significantly decreased in the case of automatic mode with manual corrections. The same group demonstrated that intra- and inter-observer variability in HNC OAR delineation was reduced when a radiation oncologist manually corrected automatic contours, with 33% of time savings (23 vs. 34 min) [60].

#### Future Prospective

In auto-contouring studies, we noticed the predominance of U-Net-based architectures that were often employed by combining multiple models in a hierarchical fashion. We emphasize that it is difficult to draw definitive conclusions on which approach was the best because the specificities of datasets, experimental context, and design make the results difficult to compare from a quantitative point of view. The current availability of public datasets and scientific challenges, such as the HECKTOR challenge [61], the StructSeg challenge (https://structseg2019.grand-challenge.org (accessed on 4 May 2023)), and the TCGA-HNSC dataset [62], provides an excellent opportunity to benchmark and compare methods. If these datasets and challenges are widely adopted by researchers, as carried out in [18,31], it would be possible to reach higher standards of evidence.

Furthermore, we highlight that only a minority of the studies evaluated the clinical impact of the proposed methods. In fact, most of them reported only topological metrics (DSC, HD95, etc.) to compare the generated contours with the ground truth. Differences from a dosimetric point of view have been evaluated only in a few studies [22,23,25]. While this is a more expensive analysis, there is no certainty that methods that offer improvements—measurable but often small—on topological metrics might have a significant clinical impact. Indeed, studies exploring the dosimetric impact of manual editing on automatic contours should recognize minor stylistic edits vs. significant edits, which might not be evident with topological metrics such as, for instance, the DSC on small volumes or for organs distant from high dose gradient regions. Therefore, we encourage future studies to be performed in a multidisciplinary arena where clinicians and computer scientists combine their expertise.

### 3.2. AI in HNC RT Planning Optimization

Treatment planning for HNC is challenging due to the non-convex geometrical shapes of the planned target volumes (PTVs) and their typical overlap with multiple OARs, requiring high beam modulations to achieve satisfactory OAR sparing. Thus, the most common delivery techniques for HNC treatments are intensity-modulated radiation therapy (IMRT) and volumetric-modulated arc therapy (VMAT). Several target volumes are treated at the same time, with doses ranging from 54 to 70 Gy administered with a standard fractionation schedule of 2 Gy/fraction, 1 fraction/day, and 5 fractions/week. Usually, simultaneous integrated boost (SIB) is used to achieve better target conformity, less dose to critical structures, moderate treatment acceleration with reduced total treatment time, and dose escalation in the GTV.

A time-consuming iterative inverse optimization procedure is required to explore the dose trade-offs between target coverage and OAR sparing to eventually obtain an optimal plan. Furthermore, the optimization outcome is heavily affected by the planner’s previous experience and the institutional practice guidelines. In recent years, several efforts have been made to develop ML and DL tools to standardize and streamline the HNC treatment planning process. The great computational power of modern processors and the predictive power of AI algorithms allowed the creation of automatic tools able to predict 3D dose distributions, plan optimization parameters, and discrepancies between planned and delivered multileaf collimator (MLC) positions. The results of the studies on AI in HNC planning are reported in Appendix A.

McIntosh et al. introduced a contextual atlas regression forest capable of predicting the dose for novel patients using a set of automatically selected most similar patients [6]. The predicted dose-per-voxel could be converted into a complete treatment plan. Compared to the clinical plans, the automatic plans achieved an average 0.6% higher dose for target coverage and an average 2.4% lower dose for the OARs.

Fan et al. developed a 3D dose prediction model whose output was used for the automatic plan generation [63]. The authors reported that statistically significant differences in dose-volume histogram (DVH) indices were found only for the brainstem, right and left lenses, and a PTV structure (PTV70.4, i.e., the highest PTV dose level). Nguyen et al. proposed a U-Net- and DenseNet-based model (HD U-Net) for dose prediction [64]. Averaging across all OARs, this model could predict the maximum dose within 6.3% and the mean dose within 5.1%. Miki et al. compared a modified filtered back projection (mFBP) they had previously developed [65] with an HD U-Net [64]. The predicted dose distribution was used to extract the optimization parameters, and the dose distribution was obtained using a commercial treatment planning system (TPS). HD U-Net achieved more accurate dose predictions than mFBP, which, however, did not require any prior learning. A DL agent for fully automated plan generation was developed by Li et al. [66]. The authors used a GAN to predict fluence maps that were automatically post-processed and imported into a commercial TPS. The authors found no significant differences between manual and DL plans for Dmean of left/right parotid glands and oral cavity and for Dmax at 0.01 cc of brainstem and cord +5 mm. Gronberg et al. discussed their participation in the Open Knowledge-Based Planning Challenge (OpenKBP) of the American Association of Physicists in Medicine [67]. The authors explored many network architectures to predict 3D dose distributions. An ensemble of three 3D Dense Dilated U-Net (3D-DDU-Net) architectures was found to be the top-performing model. The authors reported 2.6 Gy of mean absolute error between prediction and ground truth dose distributions.

A commercially available AI-based tool (DST, QuickMatch, Siris Medical) was investigated by Sher et al. to estimate the lowest achievable dose to key OARs, such as the parotid, submandibular glands, and oral cavity [68]. The authors proposed a hybrid approach where, for each OAR, the treating physician chose the lowest dose value between a physician’s custom OAR directive and the AI-driven directive predicted by the DST. The hybrid approach was found to improve the plan quality as it reduced OAR dose objectives > 3 Gy in 22% to 75% of cases.

Carlson et al. applied ML algorithms to predict discrepancies between planned and delivered movements of MLCs [69]. The model training could be performed using only a single plan due to the large amount of data available (number of leaf positions for each control point). By incorporating the predicted MLC positions into the TPS dose calculations, the authors found that the predicted dose volumetric parameters were in closer agreement with the delivered parameters than the planned parameters, particularly for OARs at the periphery of the treatment area.

Metal artifacts are common in HNC patients who are candidates for RT. A DL-based tool for metal artifact reduction (DL-MAR) was investigated by Koike et al. to evaluate its dosimetric impact compared to the conventional water density override method [70]. The authors generated synthetic artifact-free CT images from metal artifact images. The mean artifact index of the DL-MAR images was significantly smaller than that of uncorrected images (13.2 ± 4.3 vs. 267.3 ± 113.7), and greater dose differences were found between reference water plan and uncorrected than between reference and automatic correction.

Finally, Scholey et al. investigated the feasibility of MRI-only treatment planning by implementing a 3D U-Net to generate synthetic megavoltage CT (sMVCT) from paired T1-weighted MRI [71]. To assess the model’s performance in different tissue types, MVCT images were segmented into whole body, soft tissue, bone, and air-filled volumes. The highest mean absolute error (MAE) between sMVCT and MVCT averaged over all voxels was found for the bone volume (138.0 ± 43.4 Hounsfield Units [HU]), whereas the lowest DSC was observed for the air-filled structures (0.60 ± 0.11). For four representative patients, the authors compared the dose distributions calculated on MVCT and sMVCT using the gamma passing rate test, reporting values > 93% at 2%/2 mm.

#### Future Prospective

The analysis of these works reveals a certain fragmentation in the approaches used, ranging from supervised approaches (U-Net and ResNet) to adversarial ones (GAN and Cycle-GAN). In this context, the availability of datasets and/or benchmarks shared by the community might help to identify the most promising techniques, even with the problems that the definition of a benchmark would entail (insufficient variability in the data and consequent bias). Furthermore, the evaluation of the AI output is often focused on the comparison with the ground truth (i.e., the manual plan), while a real assessment of the clinical impact is, again, less frequent. Finally, a limitation of most of the approaches proposed in this area is that papers usually do not provide any measure of confidence in their predictions (a problem that also partly concerns contouring studies). Having predictions along with a confidence measure could significantly improve the effectiveness of a model from a clinical point of view.

### 3.3. AI in HNC RT Delivery

Modulated plans, such as IMRT and VMAT, allow optimization of the instantaneous beam/arc fluency for treatments as complicated as the HNC. However, the high modulation and the steep dose gradient around the target might have a dramatic impact on the delivered dose in cases of anatomic changes due to weight loss, tumor volume reduction, or positioning inaccuracies. These changes were not accounted for in the initial planning scan and should therefore be considered when assessing the actual dose received by the patient. To this end, adaptive radiotherapy (ART) has recently been introduced with the aim of treatment planning re-optimization during the course of radiation to take into account weight loss or tumor shrinkage. Thus, the radiation plan is tailored to the changing size of the tumor and normal tissue anatomy, reducing the dose to sensitive structures while minimizing dose inhomogeneity and inadequate target coverage. However, routine adoption of ART is usually limited to research studies due to temporal and logistical issues. In this context, Cone-Beam Computed Tomography (CBCT) can help identify setup errors between treatment sessions but is not suitable to precisely detect the spatial location of the tumor and normal tissues or to be used for dose re-calculation due to its inaccuracy in the reconstructed HU scale and limited field of view in the cranial-caudal direction.

Recently, several ML and DL algorithms have been investigated to post-process the daily CBCTs to automate and enhance the delivery of HNC treatments, especially for OAR segmentation and HU estimation for dose recalculation for patients treated with photons. The results of these studies on AI in HNC delivery are reported in Appendix A.

Maspero et al. adopted a 2D Cycle-GAN to facilitate dose calculation based on CBCT for HNC, lung, and breast cancer patients. Cycle-GAN could generate a synthetic CT (sCT) from an unmatched CBCT [72]. The authors reported that a single network trained on all anatomical sites achieved comparable performances with networks specifically trained for each anatomical site. Overall, image similarity was higher between sCT and rescan CT (rCT) compared to the one between CBCT and rCT, and mean dose differences < 0.5% were obtained in high-dose regions. Barateau et al. used a standard GAN to generate sCT images from CBCTs to perform the dose calculation [73]. The GAN was compared with three existing methods: (1) density to HU relation from phantom CBCT image (HU-D curve method); (2) water-air-bone density assignment method (DAM); (3) DIR. Overall, the MAE between sCT and reference CT HU values was significantly lower in the DL model than in all other methods. The authors calculated the gamma passing rate at 2%/2 mm between sCT and reference CT: the DIR achieved the best passing rate result, 98.8 ± 0.7%, while the DL model was slightly lower, with 98.1 ± 1.2%.

A semi-auto-segmentation (SAS) approach was proposed by Gan et al. to automate HNC OAR-segmentation on rCT for accurate prediction of Normal Tissue Complication Probability (NTCP) [74]. The authors employed a commercially available deep learning contouring (DLC) tool (Workflow Box 2.0, DLCExpertTM, Mirada Medical Ltd., Oxford, UK) and compared its performance to human segmentation (HS) and DIR. After the analysis, the SAS approach consisted of HS, DIR, and DLC for, respectively, the segmentation of parotid glands, pharyngeal constrictor muscle, and all the other OARs. The authors concluded that HS, especially for the parotid glands, remains necessary for accurate interpretation of the mean dose and NTCP during ART.

Chen et al. applied a 2D U-Net architecture to CBCT scans taken on the same day of treatment (oCBCT), together with input rCT scans, to produce enhanced CBCT images (eCBCT) [75]. The authors found significant improvements in mean pixel values for eCBCT OARs with improved manual contour accuracy for eCBCT-to-rCT. Furthermore, visual scoring was used to qualitatively evaluate users’ confidence in manually segmenting OARs on both eCBCT and oCBCT, showing that OAR segmentation was more accessible on eCBCT than oCBCT images.

Recently, applications of DL models to image registration tasks have been investigated. Ma et al. assessed a registration-guided DL (RgDL) framework to integrate image registration algorithms and a U-Net segmentation model [76]. Authors compared generated contours using rigid body (RB) and DIR as registration algorithms before applying the DL model, as well as using registration and DL alone. The authors reported that RgDL was less susceptible to image artifacts and that, overall, RgDL based on DIR achieved the best performance. In another study, Liang et al. proposed a method called test-time optimization (TTO) to refine a pre-trained DL-based DIR model [77]. The authors compared several DL architectures and assessed the improvement in generalization error after applying TTO. Despite finding mild average improvements in terms of DSC (0.04, 5%) and HD95 (0.98 mm, 25%), the authors reported that the improvement for outlier patients with large anatomical changes was significant. Furthermore, the proposed method could fine-tune a pre-trained model in ~3 min, thus being suitable for online ART.

Guidi et al. applied a combination of cluster analysis (K-means) and support vector machines to predict patients who would benefit from ART and re-planning interventions [78]. The ML model was trained on volume and dose variations of the parotid gland during the 6 weeks of therapy to yield the following classification: correct treatment, suggested re-planning, bias (e.g., inappropriate daily image, limited field of view, etc.), warning (abnormal variations during the treatment). The authors found that no re-planning was needed for an average of 87% of cases in the first three weeks of treatment. From the 4th week on, patients that maintained ‘‘correct treatment” decreased to 45%, while those that would benefit from re-planning increased up to 55%. During the last 2 weeks, on average, 23% were classified as "correct treatment", while 59% were classified as “suggested re-planning”.

Two recent studies by Harms et al. and Lalonde et al. investigated DL models to perform dose calculations from CBCT for adaptive proton therapy [79,80]. The first group of authors used Cycle-GAN to predict relative stopping power (RSP) maps from daily CBCT images. The authors assessed the differences between the CT-based and CBCT-based RSP maps and compared the proposed model with DIR and two other DL methods (CNN and U-Net). Overall, the proposed Cycle-GAN statistically outperformed all the other methods, and, in terms of dose comparison, the mean gamma passing rate per plan at 3%/3 mm was 94% (3 patients). The second group evaluated the performance of a 2D U-Net to perform projection-based scatter correction on CBCT images. The HU errors in scatter-free vs. scatter-corrected images were systematically lower than uncorrected vs. scatter-free. The authors also performed a 2%/2 mm gamma evaluation for 10 plans optimized on scatter-free images and recalculated on scatter-corrected images and reported an average gamma passing rate of 99%.

#### Future Prospective

Although several efforts to circumvent the technical limitations of current CBCTs are under investigation, HNC ART is still an open field of research. One shortcoming of CBCTs that was not considered in the present studies is the limited field of view. A possible solution would be to acquire an extended CBCT or two CBCT scans and combine them using image registration. Most promising is the development of automated tools for re-calculating the dose received by the targets and OARs, as well as DL-based OAR segmentation that could allow clinical re-planning on daily CBCTs. However, the lack of DL applications for automatic target segmentation and plan optimization still limits the implementation of online HNC ART in clinical practice. Once DL models are also tested on these tasks, automated and streamlined ART could become a standard practice.

## 4. Discussion

In recent years, the proliferation of new studies related to the application of AI in the field of radiation therapy has been evident. In this review, 62 original papers were critically assessed. A summary of the results found is reported in Figure 2.

A greater focus on the aspects of contouring rather than the planning, optimization, and delivery processes was observed. The segmentation of OARs was the most popular topic (35 out of 43 papers on segmentation), with the aim of reducing the time required for OAR contouring. Usually, U-Net-based architectures were adopted. The integration of manual adjustments after DL segmentation is, at present, the best choice, allowing large time savings (up to 93% [28]) compared to a complete manual procedure. We stress that studies exploring the dosimetric impact of manual editing on automatic contours should be performed to better recognize minor stylistic edits vs. significant edits. Complete automatic segmentation presented low dosimetric differences, even if reported only by a few authors [22,23,25]. Furthermore, minimal DSC improvements were found using complex models (3D multi-view [32], multiple networks [39]), with longer computation times with respect to simpler 2D models. In general, only CT-based segmentation approaches were considered. However, a few studies showed the inclusion of MRI led to better OAR delineation [46,47]. Finally, multiple imaging series were required for the target delineation, mainly PET/CT series [50,51,52], while marginal benefits were observed with MRI/CT series [55].

The majority of the planning optimization studies focused on predicting dose distributions that could be used to extract objectives and priorities for automatic optimization, reporting good agreement between AI-generated and clinical plans [6,63,65,66,68]. A limitation of these studies is that they did not provide any measure of confidence for their predictions, which could facilitate the interpretability of a model’s output in the clinic. Nonetheless, this approach could streamline a time-consuming process such as HNC planning while harmonizing the results, which are affected by a planner’s previous experience and practice. Other AI-based applications were investigated for predicting MLC leaf position errors [69], metal artifact reduction [70], and MRI-based planning [71]. Although promising, these are still in the early stages of research.

The HNC delivery process presented a concentration of adversarial methods (GAN, Cycle-GAN) to produce synthetic CTs based on daily CBCTs for dose recalculation [72,73,79,80]. Overall, gamma passing rate evaluations showed optimal agreement between sCT and CT dose distributions (>98% with 2%/2 mm), except for a study using proton therapy in which, although for only 3 patients, the mean gamma passing rate per plan at 3%/3 mm was 94% [80]. Furthermore, recent developments in DL-based image registration for OAR contour propagation have been reported, showing promising results [76,77]. However, the lack of available DL applications for automatic target segmentation and plan optimization still limits the implementation of online HNC ART in clinical practice.

Driven by the recent breakthroughs in ML and DL, AI represents a viable tool to streamline the whole RT workflow for the HNC treatment. Important advances have been made, but the need for collaborations between research centers remains evident. In the AI and computer science fields, it is common practice to define collaborative benchmarks and challenges to promote technological and scientific development, as in robotics or computer vision areas. In the RT field, such an approach would ensure more adequate sample sizes and standardization of procedures, with special emphasis on harmonization of image acquisition and storage. Nowadays, some applications of this strategy have already produced marked changes in everyday clinical practice, easing the segmentation and planning workload and supporting practitioners in reducing intra- and inter-observer variability.

However, the complete translation of AI into the clinical workflow faces certain challenges that are hindering its widespread adoption. Typically, ML/DL models require large amounts of high-quality data for their training, but accessing such datasets in healthcare can be challenging due to privacy concerns and fragmentation across institutions. Additionally, approvals from regulatory bodies are required for deploying AI systems and ensuring patient privacy, consent, and ethical use of these algorithms. More importantly, in critical domains such as RT, where decisions have a direct impact on patient outcomes, the interpretability of AI systems is crucial for clinicians to understand how AI arrives at its predictions or recommendations. Finally, integrating AI technologies seamlessly into existing clinical workflows requires overcoming technical barriers and changing established practices.

The inclusion of AI in the process of creating and delivering treatment plans should not be an "all or nothing" operation but should happen gradually and through various stages. From the perspective of a human-centered AI with the aim to amplify and augment rather than replace human abilities, the gradual integration of AI in the RT workflow could potentially better clarify its real impact in clinical practice and, moreover, pave the way for new advanced tools able to improve active collaborations between clinicians and AI researchers.

## 5. Conclusions

In this topical review, we analyzed 62 papers focusing on the role of AI in HNC RT. We found a topical polarization around the OAR segmentation process, which is currently the most mature and with commercial software nowadays available in clinical practice. We observed a general focus of the studies on the comparison between AI outcomes and ground truth, while a real assessment of the clinical impact was less frequent. Thus, we suggest that future studies be performed within interdisciplinary groups, including clinicians and computer scientists. This collaborative approach can ensure that the development and implementation of AI technologies in RT is effectively aligned with clinical needs and enhances patient care.

## Figures and Tables

**Figure 1 jpm-13-00946-f001:**
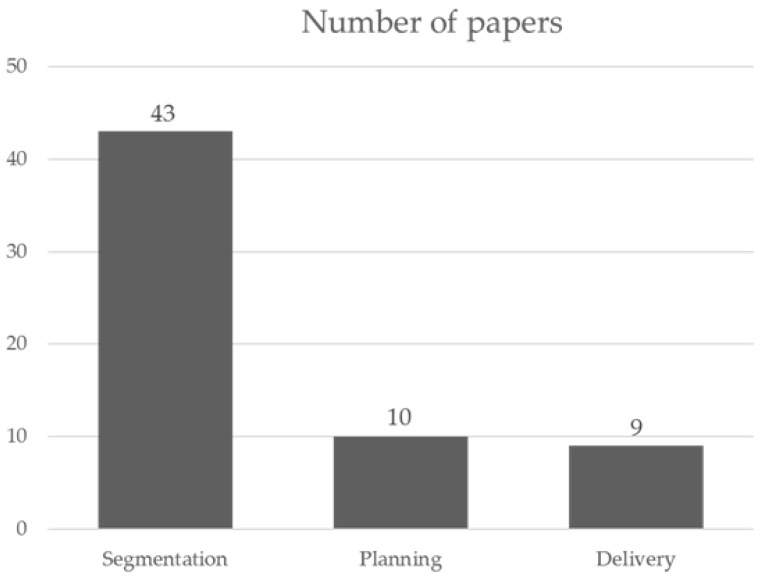
Number of papers per category selected in this review.

**Figure 2 jpm-13-00946-f002:**
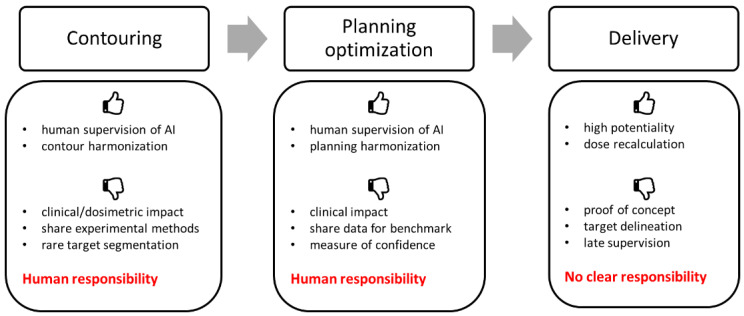
Critical summary of the studies considered in this review for each phase of the HNC RT process.

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
