# Peer review of "Enhancing Radiotherapy Workflow for Head and Neck Cancer with Artificial Intelligence: A Systematic Review"

_jpm, 2023, doi:10.3390/jpm13060946_

Round 1

Reviewer 1 Report

The article "Enhancing Radiotherapy Workflow for Head and Neck Cancer with Artificial Intelligence: A Systematic Review" proposed by Franzese et al. systematically analyzes 168 articles that address the topic of using artificial intelligence in HNC radiotherapy workflow optimization. The paper is divided into 3 symmetrically analyzed categories (a table with relevant studies for each section and a further detail of the main proposed studies. The first and the chapter considered the most represented in the literature addresses the use of AI in the automatic segmentation of OARs. The second and the 3rd chapter addresses the involvement of AI and DL/ML algorithms in the optimization of the treatment plan and respectively in the delivery of radiotherapy. I especially appreciate the mention of CBCT role in adaptive radiotherapy concepts, but also the overview of some studies related to proton therapy. As well limits are mentioned (the absence of a measure of confidence for prediction, but also an imbalance of the studies in favor of automating the contouring of OARs). However, the main remark is the absence of clinical correlations, but also the need for a transition in supervised involvement of AI in radiotherapy workflow (including human factor).The authors note the essence of a collaboration between AI and clinicians and not a future with pure substitution of the clinician in the radiotherapy workflow equation. I would also recommend the explicit mention of the reasons why AI has not yet been translated into clinical practice and I would particularize the case for radiotherapy workflow. However, I congratulate the authors for their masterful effort.

Author Response

We would like to thank the reviewer for the positive feedback. Their acknowledgement of our efforts is greatly appreciated.

We also thank the reviewer for their valuable suggestion to include the reasons why AI has not yet been fully integrated into clinical practice. We have incorporated a dedicated paragraph in the Discussion section to address this issue. Lines: 488-498

Reviewer 2 Report

Dear authors, 

I thank you for your interesting manuscript. 

It is a well written manuscript. however I have some comment / suggestion for you. 

1) you should do something about a better " data visualization". Your manuscript is more some mass text in current version. that makes it somehow difficult to follow and less interesting for readers. 

2) maybe you can more explain and better define about your selection criteria in search strategy. A selection method just by 2 or more experts without a well defined criteria is more prone for bias to select relevant references. 

3) Generally is a topical review much shorter than your manuscript. As reason, You repeat the result of many articles in your text and regarding to comment 1 maybe you can solve it by better presentation to make it shorter and more interesting. 

Author Response

We thank the reviewer for the positive feedback and the comments which improved the paper quality.

1) Many thanks for this observation. To reduce the volume of text, we have moved the Tables describing the selected papers in the supplementary material. Additionally, to better visualize the number of papers selected per group, we have added a graph showing this statistic in Figure 1.

2) We understand the reviewer’s concern regarding the potential bias that may arise in the selection process. Our criteria encompassed relevance to the topic, novelty, methodology, and quality of evidence. These criteria were followed to minimize subjective judgments. Moreover, by involving multiple reviewers, we aimed to reduce individual biases that could potentially influence the selection process. We have better specified our selection criteria in the Materials and Methods section. Lines: 78-79.

3) We have made several changes throughout the manuscript to reduce its length and make it more concise. By moving the tables in the supplementary material, we think now the text provides comprehensive results while maintaining a concise narrative.

Round 2

Reviewer 2 Report

Dear authors, 

I thank you for your effort to revised your manuscript. I have no other comments. 

Author Response

We thank again the reviewer for their comments which improved the paper quality.